# Psychological Reactions during and after a Lockdown: Self-Efficacy as a Protective Factor of Mental Health

**DOI:** 10.3390/ijerph20176679

**Published:** 2023-08-29

**Authors:** Francesco Ruotolo, Gennaro Ruggiero, Zaira Cattaneo, Maria Arioli, Michela Candini, Francesca Frassinetti, Francesca Pazzaglia, Ferdinando Fornara, Andrea Bosco, Tina Iachini

**Affiliations:** 1Department of Psychology, Università degli Studi della Campania “L. Vanvitelli”, 81100 Caserta, Italy; gennaro.ruggiero@unicampania.it (G.R.); santa.iachini@unicampania.it (T.I.); 2Department of Social and Human Sciences, University of Bergamo, 24129 Bergamo, Italy; zaira.cattaneo@unibg.it (Z.C.); maria.arioli@unibg.it (M.A.); 3Department of Psychology, University of Bologna, 40126 Bologna, Italy; michela.candini2@unibo.it (M.C.);; 4Department of General Psychology, University of Padua, 35121 Padua, Italy; francesca.pazzaglia@unipd.it; 5Department of Pedagogy, Psychology, Philosophy, University of Cagliari, 09124 Cagliari, Italy; ffornara@unica.it; 6Department of Educational Sciences, Psychology, Communication, University of Bari “Aldo Moro”, 70121 Bari, Italy; andrea.bosco@uniba.it

**Keywords:** lockdown, mental health, anxiety, stress, self-efficacy, isolation, COVID-19

## Abstract

The aim of the present study was to investigate the effects of home confinement/social isolation (i.e., lockdown), imposed to reduce large-scale spread of a disease in the population, on the mental health of individuals. Through an online survey during the lockdown (*DL*) related to COVID-19 (1085 respondents, 627 females, age_range_: 18–82) (Italy, 23 April–2 May 2020), we revealed that situational factors, i.e., the presence of children at home and female gender, and psychological factors, i.e., a greater sense of isolation, lower perception of safety outside the home and higher trait anxiety, predicted higher levels of state anxiety (R^2^ = 0.58). The same factors, but with young age instead of the presence of children, predicted higher levels of perceived stress (R^2^ = 0.63). Then, these data were compared with those collected after the lockdown (*AL*) (174 respondents, 128 females, age_range_: 19–78) (Italy, 1 July–31 October 2021). The results showed that along with a reduced sense of isolation (*DL* = 2.90 vs. *AL* = 2.10) and an increased perception of safety outside the home (*DL* = 2.63 vs. *AL* = 3.05), a reduction in state anxiety (*DL* = 45.76 vs. *AL*= 40.88) and stress appeared (*DL* = 18.84 vs. *AL* = 17.63). However, the situation was better for men than for women. Perceived self-efficacy emerged as a protective factor for mental health (R^2^_range_: 0.03–0.27). The results are discussed in light of the evidence on the effects of lockdown on individuals worldwide. These results may be used to make more educated decisions on targeted help for individuals who may be most adversely affected by the adoption of lockdowns in the future.

## 1. Introduction

At the beginning of April 2020, almost 3.5 billion people worldwide were asked to stay confined to their homes and reduce social contacts to a minimum in order to contain the spread of COVID-19, a disease found in China and then globally in late 2019 and early 2020 [1]. As such, this strategy has been effective. However, it has led to an increase in mental health issues and a deterioration in well-being [2,3,4,5,6,7,8,9,10,11].

The term “lockdown” has been used to indicate this forced isolation. It is often a harsh and unpleasant experience for people. In fact, it involves being confined at home, with extreme limitations on the live contact with loved ones, going out, and moments of leisure outside, and is accompanied by a profound level of uncertainty about the evolution of the disease and the contagion [6]. For those subject to this restrictive measure, the impact can be dramatic. For example, studies about the effects of lockdown related to epidemics/pandemics prior to COVID-19 (e.g., plague and cholera) have reported cases of suicide [12], and increases in the level of public anger, resulting in an increase in lawsuits [13]. Moreover, depression, stress and post-traumatic stress symptoms have also been widely documented (for a review of lockdown effects on individuals before 2020 see [14]). The situation was no better for the more recent 2020 lockdown. In fact, several studies published both after [2,3,4,5,6,7,8,9,10,11] and during [15,16,17,18,19,20,21,22,23,24,25,26] the pandemic about the impact of lockdown on people’s mental health have reported increases in anxiety, depression, and post-traumatic stress symptoms among confined individuals. According to Hawkley and Cacioppo [27], humans have learned to cooperate with each other to survive in hostile environments and adverse times. Social bonds, especially with family and friends, fostered these cooperative and affiliative behaviors. Genetic, neural and hormonal changes have supported these behaviors throughout evolution. Because our sense of connection to others is embodied in the physical organism, it makes us deeply social organisms and serves as a scaffolding for our physical and mental well-being and integrity. When this scaffolding fails or is damaged, “the rest of the self begins to crumble” ([27] p. 219).

It has been shown that the effect of a lockdown on people’s mental health may depend on several factors. For example, studies carried out before 2020 showed that significant predictors of post-traumatic stress among individuals subjected to a lockdown were excessive worry about being infected, feeling isolated, fear of running out of food and/or inadequate housing, and limited or no information about the spread of infection [14]. Similarly, more recent surveys carried out among individuals confined at home due to COVID-19 have revealed that their level of anxiety increased the more they followed the news about the spread and impact of the COVID-19 virus [28]. Moreover, young people (21–40 years) [5,11], especially women with children at home [4,7,9,29], showed higher levels of anxiety, depression and stress. Finally, high levels of anxiety were found in people with relatives or friends with COVID-19 disease [7], and in people who had a history of medical problems and poor health [3,4,11]. In sum, these studies revealed that both psychological and situational factors might modulate the effects of lockdown on an individual’s mental health.

Since high levels of stress and anxiety are well-known risk factors for various psychopathologies and alter the individual’s immune system [30], it is of fundamental importance to continue investigating the possible predictors of stress and anxiety resulting from lockdowns imposed during a pandemic or other emergency situations [11]. More importantly, most of the studies carried out during the pandemic are cross-sectional [14,15,16,17,18,19,20,21,22,23,24,25,26] and do not take into account the temporal dimension of confinement. In this regard, two aspects are crucial: the duration of the lockdown, and what happens after the confinement. Very little data is currently available on the mental health of individuals once the restrictions have ended or on the relationship between mental health and days of lockdown [6]. Notably, according to some studies, high levels of self-efficacy play a key role in preserving the mental health of individuals during lockdown [31,32,33,34]. Self-efficacy is defined as the perception of one’s own ability to succeed in specific situations or accomplish a task [35,36]. For this reason, it is also important to examine whether self-efficacy plays a protective role in recovering mental well-being after a lockdown.

Therefore, the aims of the current study were (i) to investigate which psychological and situational factors predicted the perceived stress and anxiety among individuals subjected to the lockdown to counter the COVID-19 outbreak in Italy; (ii) to review the mental health status of individuals once the restrictive measures were over; and (iii) to assess the role of self-efficacy in predicting mental health. With these aims, state anxiety and perceived stress levels of a sample of adults were measured through two online surveys: the first *during* the lockdown (between 23 April and 2 May 2020), and the second *after* the lockdown (between 1 July and 31 October 2021). Furthermore, regarding the psychological factors, the perceived sense of safety both inside and outside home along with more commonly studied predictors, such as feeling isolated and trait anxiety levels, were measured [37,38,39]. In regard to the situational factors, the absence of a stable partner (i.e., singleness), the context in which one lives (i.e., city, town or country), presence of children, confinement days, and number of outings were considered [11]. In addition, the respondents’ sex and age were also taken into account. Finally, we also measured the level of self-efficacy of the participants after the end of the restrictive measures.

Based on previous studies [2,3,4,5,6,7,8,9,10,11,14,15,16,17,18,19,20,21,22,23,24,25,26], we put forward the following hypotheses:

**H_1_:** 
*A significant relationship between psychological and situational factors and state anxiety levels should emerge. Specifically, higher levels of state anxiety would be predicted by being a woman, young and single, having children at home and living in a city. Moreover, we expected higher levels of state anxiety to be associated with higher feelings of isolation and trait anxiety, a lower feeling of security (more outside than inside the home), a reduced number of outings and a longer duration of lockdown;*


**H_2_:** 
*If perceived stress is predicted by the same factors as anxiety, then we would expect the same associations as in H_1_. Importantly, the analysis will also allow us to identify common and, if any, different predictors of anxiety and stress;*


**H_3_:** 
*If the lockdown results in high levels of stress and anxiety, an increased feeling of isolation and a reduced feeling of security outside the home, then after the lockdown a decrease in state anxiety, stress and feeling of isolation and an increased feeling of security outside the home should be observed. However, the overall picture is likely to be worse for women than for men;*


**H_4_:** 
*If self-efficacy plays a protective role in recovering mental well-being after a lockdown, then a significant relationship between self-efficacy and stress, anxiety, feelings of isolation and security should be observed. Specifically, higher levels of self-efficacy should be associated with lower levels of stress, state anxiety and isolation, and higher levels of a sense of safety outside the home.*


## 2. Materials and Methods

### 2.1. Participants

First survey: The sample size was estimated with G*Power 3.1 [40]. The α was set to 0.05 and Power to 0.80. This analysis indicated a total of 837 participants would be necessary to detect a small effect size (0.02) with 11 predictors (the effect size was taken from a previous study [41]. Data collection was carried out during the first lockdown imposed in Italy to counter the spread of COVID-19. It began on 23 April 2020 and ended on 2 May 2020 when the minimum number of participants indicated by G*Power had been exceeded. The universities that collaborated with data collection were the University of Campania, University of Bologna, University of Bergamo, University of Bari, University of Cagliari, and University of Padova. The final sample comprised 1085 respondents (627 females and 458 males), aged 18 to 82 years (M = 38.78, SD = 16.08). The percentage of participants for each Italian region was as follows: Lombardy = 17%, Emilia-Romagna = 19%, Veneto = 15%, Campania = 23%, Apulia = 10%, Sardinia = 11%, and Other regions = 5%. The participants were recruited through social media announcements on the Internet, word of mouth and e-mail lists from the laboratories of the universities involved. Informed consent was obtained from all participants. The link to participate in the online survey was directly available in the posts and emails sent out. At the end of the survey, participants were asked for permission to be contacted again for possible follow-up of the study.

Second survey: For the second survey, participants from the first survey were contacted again and a total of 174 participants agreed to complete the survey (Females = 128, Males = 46; age range: 19–78, M = 31.72, SD = 14.07). The survey was conducted once the lockdown and the more severe restrictions were over, i.e., from 1 July to 31 October 2021. A sensitivity analysis was carried out to identify the minimum effect size that could be reliably detected with 174 participants, with α = 0.05 and Power = 0.80. The results showed that for the ANOVAs, the minimum effect size was 0.14 (Cohen’s f). Regarding regression, the minimum effect size was 0.04. Similar to the first survey, the sample composition of the second survey was as follows: Lombardy = 17%, Emilia-Romagna = 10%, Veneto = 14%, Campania = 34%, Apulia = 11%, Sardinia = 10%, and Other regions = 4%.

The recruitment and testing were conducted according to the ethical standards set by the Declaration of Helsinki (2013) and the Institutional Review Board of the Department of Psychology (University of Campania; Protocol no. 16.20). Informed consent was obtained from all the participants.

### 2.2. Materials

#### 2.2.1. Psychological Factors

State-Trait Anxiety Inventory (STAI): The STAI [42,43] measures both the level of anxiety present at the time of assessment (i.e., state anxiety) and the genetic predisposition to be anxious (i.e., trait anxiety). The latter is more of a personal trait. It is a self-reported inventory with a 4-point Likert scale in which 20 items (S scale) measure feelings of worry, nervousness, apprehension, tension, etc. at the moment (i.e., “How do you feel now?”), while another 20 items (T scale) measure relatively stable aspects of “anxiety vulnerability”, such as general calmness, confidence and security. The scores range from 20 to 80, with higher scores corresponding to greater anxiety. To detect significant clinical symptoms on the S-Anxiety scale, a cut-off of 39–40 is indicated [44,45,46]. Some studies suggest a higher cut-off of 54–55 [43]. In general, several studies have shown both acceptable internal consistency (Cronbach’s alpha of 0.91 to 0.95 for the S scale; 0.85 to 0.90 for the T scale) and good test–retest reliability (0.49 for the S scale and 0.82 for the T scale) [47].Perceived Stress Scale (PSS): The PSS was used to assess the perceived stress levels [48,49,50]. It measures the extent to which a person perceives that the demands of everyday life situations exceed their ability to cope with them, i.e., how stressful they are. The instrument consists of 10 items rated on a 5-point Likert scale ranging from 0 = ‘never’ to 4 = ‘very often’. Participants are asked to indicate the extent to which they have felt or thought a certain way in response to stressful situations in the past month. Scores range from 0 to 40, with higher scores indicating higher perceived stress (low stress: between 0 and 13; moderate stress: between 14 and 26; high stress: between 27 and 40). A satisfactory internal consistency (overall Cronbach’s alpha was 0.74) and test–retest reliability (i.e., 0.86 after one week, 0.61 after one year) of the PSS has been observed [51,52,53].General Self-Efficacy Scale (GSES): The GSES measures how well a person feels able to cope with a range of difficult demands in life [54]. The instrument consists of 10 items scored on a 4-point Likert scale. The items refer explicitly to personal agency, i.e., believing that one’s own actions are responsible for achieving positive outcomes. Scholz and colleagues [55] reported high reliability of the scale (alpha range: 0.75–0.92, across 25 different countries) and its validity has been shown in several works (see: http://userpage.fu-berlin.de/~health/faq_gse.pdf accessed on 21 August 2023).

An ad hoc self-report questionnaire with a five-point Likert-type scale (1 = not at all, 5 = very much) was used to assess other psychological reactions to the experience of lockdown (see [41]). The questions were (a) How isolated do you feel? (b) How safe do you feel at home? and (c) How safe do you feel outside home?

#### 2.2.2. Situational Factors

The respondents reported how many days they were home due to government restrictions (open response) and how often they left home on average (never, rarely—twice per month, sometimes—once per week, often—many times per week, always—almost daily). The interviewee was also asked to indicate his or her place of residence (i.e., city, village, country), marital status (single or not) and whether he or she had children.

### 2.3. Procedure

PsyToolkit was used to conduct the online survey [56,57]. In both the first and second survey, once participants clicked on the link received by e-mail or found on social media on the Internet, they were shown the informed consent page and the instructions. Participants were able to begin the survey after reading the instructions and digitally signing the consent form. Participants first provided demographic information (i.e., gender and age), then completed the situational factors of lockdown section, followed by the psychological reactions to the lockdown experience section, the perceived stress scale, and the anxiety scale. In the second survey (i.e., after the confinement), the General Self-Efficacy Scale was added. The time required to complete the survey was 20–25 min.

### 2.4. Data Analysis

Hypothesis 1 (H_1_) was tested through a multiple regression analysis with ‘state anxiety’ as the criterion variable. The predictors were age, gender (1 M, 0 F), being single (1 yes, 0 no), children at home (1 yes, 0 no), context (1 city, 2 village, 3 country), number of restriction days, number of outings, perceived safety at home, perceived safety outside home, feeling of isolation, and trait anxiety;

Hypothesis 2 (H_2_) was tested through a multiple regression analysis with ‘perceived stress’ as the criterion variable. The same predictors were used as for H_1_;

Hypothesis 3 (H_3_) was tested through ANOVAs for mixed design with Gender as a variable between two-levels and Time as a variable within each level (During vs. After lockdown). The dependent variables were: Anxiety levels, Perceived Stress, Feeling of Isolation and Feeling of Security during and after the lockdown. Anxiety and Feeling of Security were also considered as two-level factors, i.e., State vs. Trait Anxiety, and Safety Inside vs. Outside the home. A Bonferroni test was used for the multiple comparisons;

Hypothesis 4 (H_4_) was tested through a multivariate regression with self-efficacy as the predictor and Feeling of Isolation, Safety Inside and Outside the Home, state Anxiety and Perceived Stress as outcome variables.

## 3. Results

### 3.1. What Are the Predictors of State Anxiety during Confinement (H_1_)?

The multiple regression analysis using a backward stepwise procedure revealed a model with five predictors: F(5, 1079) = 295.59, *p* < 0.0001, R = 0.76, R^2^ = 0.58. As shown in Table 1, the predictors Gender, Children, Safety Out, Isolation and Trait Anxiety contributed significantly to the model: higher Feelings of Isolation, higher Trait Anxiety, being female and having children at home all predicted higher state anxiety, whereas feeling safer outside the home predicted lower state anxiety.

### 3.2. What Are the Predictors of Perceived Stress during Confinement (H_2_)?

The multiple regression analysis with the backward stepwise procedure revealed a model with five predictors: F(5, 1079) = 361.28, *p* < 0.0001, R = 0.79, R^2^ = 0.63. As shown in Table 2, the predictors Age, Gender, Safety Out, Isolation and Trait Anxiety contributed significantly to the model: higher Feelings of Isolation, higher Trait Anxiety, and being female all predicted higher perceived stress, whereas feeling safer outside of the home and a greater age predicted lower perceived stress.

### 3.3. Does the End of the Lockdown Lead to Improved Mental Health (H_3_)?

#### 3.3.1. Anxiety Levels during and after Confinement

A main effect of Gender emerged: F(1, 172) = 11.02, *p* < 0.0001, η^2^_p_ = 0.06. Females reported higher levels of anxiety (M = 45.86, SE = 0.84) than male participants (M = 40.40, SE = 1.41). Finally, the significant interaction between Time and Anxiety (F(1, 172) = 30.63, *p* < 0.0001, η^2^_p_ = 0.15) revealed lower levels of State Anxiety after than during the lockdown, but no significant differences emerged for Trait Anxiety (see Table 3a and Figure 1). In addition, State Anxiety was higher than Trait Anxiety both during (*p* < 0.001) and after confinement (*p* < 0.016).

#### 3.3.2. Perceived Stress during and after Confinement

The results showed that females reported higher stress levels (M = 20.12; SE = 0.53) than male participants (M = 16.40; SE = 0.89) (F(1, 172) = 12.25, *p* < 0.001, η^2^_p_ = 0.07). Moreover, stress levels reported during confinement were higher than those reported after confinement, even if this difference only approached statistical significance: F(1, 172) = 3.73, *p* = 0.055, η^2^_p_ = 0.02 (see Table 3b).

#### 3.3.3. Feeling of Isolation during and after Confinement

The results showed that females reported feeling overall more isolated (M = 2.68; SE = 0.07) than male participants (M = 2.28; SE = 0.11) (F(1, 172) = 12.45, *p* < 0.001, η^2^_p_ = 0.08). Moreover, the feeling of isolation reported during confinement was higher than that reported after confinement (F(1, 172) = 52.41, *p* < 0.0001, η^2^_p_ = 0.23) (see Table 3c).

#### 3.3.4. Safety Inside and Outside the Home during and after Confinement

The results showed that female participants reported feeling less safe (M = 3.48; SE = 0.06) than male participants (M = 3.69; SE = 0.03) (F(1, 172) = 8.65, *p* < 0.005, η^2^_p_ = 0.05). Importantly, the significant interaction between feeling of safety inside/outside the home and time (F(1, 172) = 17.66, *p* < 0.0001, η^2^_p_ = 0.09) revealed that the feeling of safety outside the home increased after confinement, while the feeling of safety inside the home was always high and did not change (see Table 3d and Figure 2). In addition, the feeling of safety at home was higher than the feeling of safety outside the home, both during and after confinement (at least *p* < 0.0001).

### 3.4. Does Self-Efficacy Predict the Mental Health of Individuals after Confinement (H_4_)?

The results revealed that self-efficacy predicted Feeling of Isolation (R^2^ = 0.04, F(1, 172) = 8.21, *p* < 0.01) and of Feeling of Security Outside the Home (R^2^ = 0.03, F(1, 172) = 6.99, *p* < 0.01), as well as, to a greater extent, State Anxiety (R^2^ = 0.24, F(1, 172) = 54.74, *p* < 0.0001) and Perceived Stress (R^2^ = 0.27, F(1, 172) = 33.79, *p* < 0.0001). Conversely, self-efficacy did not significantly predict the Feeling of Security Inside the Home (R^2^ = 0.006, F < 1). Specifically, the higher the self-efficacy, the lower the Feeling of Isolation, Perceived Stress and State Anxiety and the higher the Feeling of Security Outside the Home (see Table 4).

## 4. Discussion

The current study aimed to investigate the psychological (i.e., feeling of isolation, feeling of security inside and outside the home, and trait anxiety) and situational (i.e., age, gender, marital status, presence of children at home, context of residence, number of days of confinement, number of outings) predictors of perceived stress and anxiety among individuals subjected to the lockdown to counter the COVID-19 outbreak in Italy. More importantly, the temporal aspect of the lockdown was considered by assessing individuals’ stress and anxiety levels both during (2020) and after (2021) the COVID-19-related restrictions. Finally, we tested whether self-efficacy could be a protective factor for mental health recovery once the restrictions were over.

### 4.1. Predictors of State Anxiety and Perceived Stress during the COVID-19 Lockdown

Studies generally show an increase in stress and anxiety levels in the population during a crisis [58,59]. Consistent with this, we found that about 70% of the sample reported high levels of state anxiety (above the cut-off of 39; and about 25% above the cut-off of 54), and about 84% reported moderate (score range: 16–25) or even high (16%, score range: 26–40) levels of perceived stress.

It is well known that both high levels of anxiety and stress can have long-term effects on the body and mind. Common consequences include gastrointestinal problems and chronic heart disease, worsening headaches and migraines, sleep problems and depressive states of different clinical severities [60,61]. Although both high levels of anxiety and stress can have similar negative effects on psychophysical well-being, there is a subtle difference between the two. Anxiety is characterized by an excessive worry that may linger even in the absence of a specific menacing element. Stress is typically caused by an external trigger, even of short duration. In line with our hypotheses (H_1_ and H_2_), results from the current study revealed that both stress and anxiety share common predictors, such as an individual’s “trait anxiety” and “gender”, feeling of “isolation” and feeling of “safety outside”. However, “age” only predicts stress, and the presence of “children” to care for only predicts anxiety.

Regarding common predictors, the results showed that the higher the trait anxiety, the higher the level of state anxiety and perceived stress. This result confirms that forced isolation has extremely negative repercussions for the most psychologically fragile individuals [62], especially for those who are likely to experience fear and worry in a variety of situations [63]. Furthermore, as hypothesized, the more people felt less safe outside the home, the more they reported high levels of state anxiety and stress. In contrast, the sense of security at home did not seem to predict mental health issues. This confirms that people perceived contact with other individuals outside as potentially dangerous. While this fear may have positive consequences for combatting the spread of the virus, it has negative consequences in terms of social isolation. In fact, a high feeling of isolation was associated with high levels of anxiety and stress. It has been shown that during COVID-19-related lockdowns, a large segment of the population experienced a significant increase in self-reported loneliness [64]. This is worrisome, as loneliness is associated with substance use and accelerated cognitive decline [27], as well as a significantly elevated suicide-related mortality rate [64,65].

Finally, female participants indicated higher levels of stress and state anxiety than male participants. Adams-Prassl and colleagues [10] found that in the US, restrictions of freedom due to COVID-19 reduced “*mental health by 0.083 standard deviations*” and that “*this large negative effect was driven entirely by women*” (see also [66,67,68]). According to the Organization for Economic Co-operation and Development and the European Parliament, women have been at the forefront of COVID-19. This is mainly because more women than men work in health care systems. In addition, women often do most of the unpaid family care work, and thus face higher risks of being dismissed and/or exposed to higher levels of abuse or harassment during a lockdown (Women at the core of the fight against COVID-19 crisis (oecd.org); Understanding the impact of COVID-19 on women (infographics)|News|European Parliament (europa.eu)) [69,70]. Unfortunately, this adds to a body of evidence showing that acute psychological disorders seem to be more prevalent in women than in men following adverse or traumatic events [71,72,73,74,75,76].

Regarding the different predictors for anxiety and stress, the presence of children at home was associated with higher levels of anxiety. Di Giorgio et al. [77] found more sadness and frustration in a group of 245 Italian mothers during the COVID-19 lockdown than before the pandemic. Similarly, Benassi et al. [78] found higher rates of moderate and severe anxiety in a group of women with children compared with those without children. Thus, our findings are consistent with previous evidence suggesting a greater increase in emotional disturbances and exhaustion, low mood and irritability in parents compared with non-parents in lockdown [79]. It is plausible that this can be attributed to difficulties in balancing personal life, work/smart-working and childcare. School closures worsened the situation as parents had to deal with their children’s education and learning [80]. Moreover, increased anxiety in parents during a forced lockdown may also be associated with a state of constant worry about the health and future of their children [29].

Partly in line with our hypotheses, the age of the participants predicted perceived stress but not anxiety levels. Specifically, as age increased, stress decreased, thus revealing that the population most affected by confinement was the young compared with the elderly. This result may seem counterintuitive. In fact, COVID-19 infections as well as deaths predominantly affected the population over 50 years old and the elderly in frail health. It was precisely to them that precautions were most targeted and social isolation recommended. Actually, our finding is in line with previous studies [81] that reported higher scores for stress, anxiety, and depressive symptoms in younger people than in older people. How can this be explained? On the one hand, elderly people tend to be more sedentary than younger people (see [81]). In addition, people over 60 are also more likely to have experienced epidemics or pandemics in the past, from which they may have acquired effective coping strategies. In contrast, younger people may have been more concerned about COVID-19′s threat to their academic, social, employment and economic prospects [5,82,83].

Contrary to our expectations, the number of days of restriction and the place of residence (i.e., country, village or city) did not predict the mental health of individuals. This may be an indication that negative events can have traumatic effects regardless of their duration and where people live, or that negative effects may occur after a certain threshold.

### 4.2. Mental Health before and after Lockdown and Gender Differences

As hypothesized (H_3_), the results of the survey conducted after the end of the COVID-19 restrictions showed a significant reduction in state anxiety and, to a lesser extent, in perceived stress (in the latter case, the difference ‘during vs. after’ only approached statistical significance). Similarly, the perception of safety outside the home had increased compared with a year earlier, and the feeling of loneliness had reduced. This clearly indicates that isolation put a strain on individuals’ mental health and that, once isolation is over, resuming social life can help people recover. However, these encouraging findings show a worse picture for women than for men, even after the end of the lockdown. This is probably related to the fact that women tend to suffer from post-traumatic stress disorder to a greater extent than men [84,85,86].

### 4.3. The Role of Self-Efficacy as a Protective Factor of Mental Health

Interestingly, the results of the second survey revealed that a protective factor after the end of restriction is perceived self-efficacy. Perceived self-efficacy refers to an individual’s ability to perceive his or her own behavior/actions as appropriate for dealing with difficult situations. This is precisely why self-efficacy is associated with mental health and psychological disorders [87,88]. It also acts as a mediator in the relationship between stressful events and depression [89]. Specifically, Schönfeld and colleagues [90] found that self-efficacy acts as “*a mediator between the effect of everyday stress on positive and negative mental health, including symptoms of depression, anxiety, and stress*” (p. 7). In short, high levels of self-efficacy are associated with low levels of anxiety. Furthermore, several studies have shown that self-efficacy acted as a protective factor for mental health during lockdown periods [31,32,33,34,66,91]. In addition, self-efficacy proves crucial in overcoming particularly traumatic events [92,93]. For example, Poole et al. [94] showed that individuals with low levels of self-efficacy, compared with those with high levels of self-efficacy, had stronger emotional dysregulation following adverse experiences.

### 4.4. Limitations of the Current Study

In contrast to the high level of cooperation in the first survey, many respondents were no longer available in the second survey and others, having returned to their pre-pandemic lives, had less time to complete the questionnaires. This resulted in high drop-out rates and longer data collection times in the second survey than in the first. In addition, the sample of the second survey had a larger number of women than men compared with the first survey. This may limit the clarity of the comparison between the two different times. Another limitation is the small number of measures used to assess the mental health and psycho-physical well-being of the respondents. For example, many of the respondents may have been affected by COVID-19, and many others may have mental problems in addition to anxiety and perceived stress, such as depression and post-traumatic stress disorder. The inclusion of these measures would certainly have provided a more complete picture of the effects of lockdown on individuals’ mental health. Finally, the type of sampling used for data collection prevents the possibility of testing hypotheses and/or drawing specific conclusions about the respondents’ socio-economic status [9]. In fact, the most disadvantaged sectors of the population were not reached by our survey.

## 5. Conclusions

Over the centuries, the adoption of lockdowns has proven to be an effective measure not only to stem the spread of disease and the emergence of pandemics (see the recent COVID-19), but in general, to deal with perceived emergency situations. However, there may be negative consequences on the mental health of individuals [2,3,4,5,6,7,8,9,10,11]. In line with this, the current study reports the following findings:(i)Females showed higher levels of trait anxiety and feelings of isolation, which were associated with higher levels of perceived stress and state anxiety *during* lockdown. Interestingly, the presence of children at home was associated with higher levels of anxiety but not stress, whereas young age was associated with higher levels of stress but not anxiety.(ii)More importantly, the novelty of this study lies in having compared the picture that emerged during confinement with that after the end of restrictions. *After* the lockdown, levels of stress, state anxiety and feelings of isolation had decreased, and the sense of safety outside the home had increased, but this was more true for men than for women.(iii)Finally, the current study crucially showed that self-efficacy may play a protective role in recovering mental well-being *after* a lockdown. In fact, higher levels of self-efficacy were associated with lower levels of perceived stress, state anxiety and sense of isolation, and with higher levels of perceived safety outside home.

In sum, the results of this study contribute to understanding about the common and different predictors of anxiety and stress and which segments of the population are most affected, both during and after lockdowns. This information should be used to implement lockdowns in ways that try to mitigate their negative consequences, especially for women and young people. In this respect, the current study emphasizes the key role played by self-efficacy. Specifically, it would be important that government decision makers adopt measures and/or communication strategies that support the individual’s perceived self-efficacy. Therefore, future studies should investigate and test forms of communication and activities to be proposed to individuals during a lockdown to increase their self-efficacy. Furthermore, further studies should investigate whether self-efficacy is effective as a protective factor also in the more economically disadvantaged portion of the population [9] and/or among those who have unsatisfactory housing solutions for confinement [95].

## Figures and Tables

**Figure 1 ijerph-20-06679-f001:**
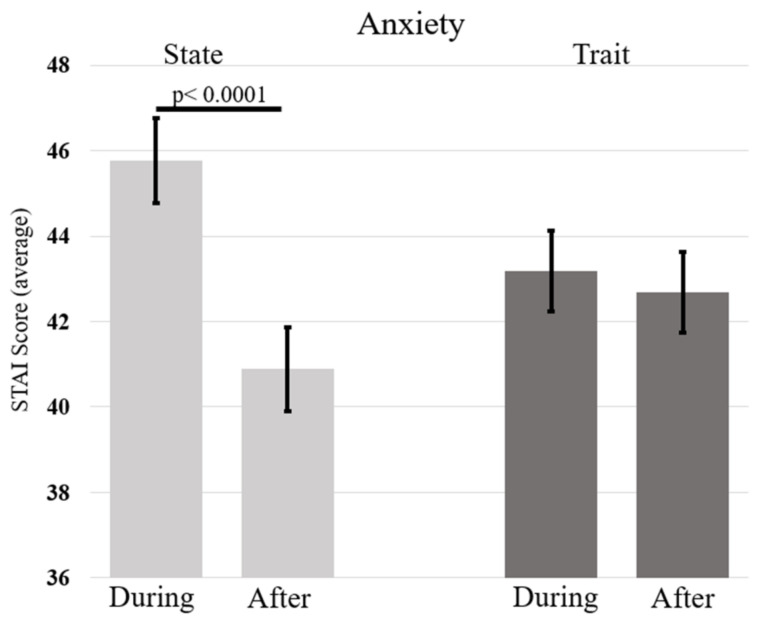
The figure shows the average state and trait anxiety levels as a function of the time of lockdown, i.e., during and after. Vertical bars represent standard errors.

**Figure 2 ijerph-20-06679-f002:**
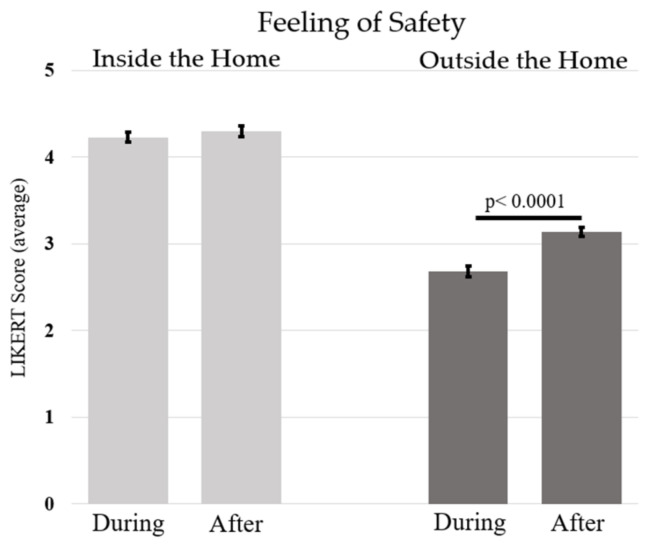
The figure shows the average feeling of safety as a function of the time of lockdown, i.e., during and after. Vertical bars represent standard errors.

**Table 1 ijerph-20-06679-t001:** Predictors of State Anxiety during confinement. * *p* < 0.001.

	*B (Stand.)*	*St. Err*	*B*	*t*	*p*
Gender	−0.09	0.02	−2.11	−4.57	*
Children	0.10	0.02	2.27	4.71	*
Safety Out	−0.15	0.02	−2.17	−7.61	*
Isolation	0.23	0.02	2.43	10.89	*
Trait Anxiety	0.60	0.02	0.63	27.21	*

**Table 2 ijerph-20-06679-t002:** Predictors of Perceived Stress during confinement. * *p* < 0.001.

	*B (Stand.)*	*St. Err*	*B*	*t*	*p*
Age	−0.11	0.02	−0.05	−5.64	*
Gender	−0.08	0.02	−1.22	−4.34	*
Safety Out	−0.07	0.02	−0.68	−3.91	*
Isolation	0.12	0.02	0.84	6.22	*
Trait Anxiety	0.66	0.02	0.46	32.11	*

**Table 3 ijerph-20-06679-t003:** Psychological measures during and after lockdown and gender differences: descriptive statistics (mean and standard error).

		Time	Mean (SE)	Sex	Mean (SE)
(a)	State Anxiety	During	45.77 (1.00) *^1^	Male	41.65 (1.72)
				Female	49.88 (1.03)
		After	40.89 (0.98) *	Male	38.50 (1.67)
				Female	43.27 (1.00)
	Trait Anxiety	During	43.18 (0.95) ***	Male	40.52 (1.63)
				Female	45.84 (0.98)
		After	42.68 (0.94) ***	Male	40.91 (1.61)
				Female	44.45 (0.97)
(b)	Perceived Stress	During	18.85 (0.64) **	Male	16.48 (1.09)
				Female	21.22 (0.66)
		After	17.64 (0.58) **	Male	16.33 (0.99)
				Female	18.95 (0.59)
(c)	Feeling of Isolation	During	2.90 (0.09) *	Male	2.63 (0.16)
				Female	3.16 (0.09)
		After	2.07 (0.09) *	Male	1.93 (0.15)
				Female	2.20 (0.09)
(d)	Security Inside the Home	During	4.23 (0.06) ***	Male	4.28 (0.10)
				Female	4.17 (0.06)
		After	4.30 (0.06) ***	Male	4.37 (0.10)
				Female	4.23 (0.06)
	Security Outside the Home	During	2.68 (0.06) *	Male	2.78 (0.11)
				Female	2.58 (0.07)
		After	3.14 (0.06) *	Male	3.33 (0.09)
				Female	2.95 (0.06)

^1^ The * indicates the *p*-value associated with the statistical difference between, during and after lockdown: * *p* < 0.0001; ** *p* = 0.055; *** *p* > 0.10.

**Table 4 ijerph-20-06679-t004:** The predictive role of self-efficacy. * *p* < 0.001, ** *p* < 0.05.

		*B (Stand.)*	*St. Err*	*B*	*t*	*p*
Self-efficacy	Isolation	−0.03	0.01	−0.21	−2.87	*
	Safety Out	0.02	0.01	0.20	2.64	**
	State Anxiety	−0.88	0.12	−0.49	−7.38	*
	Stress	−0.55	0.07	−0.52	−8.03	*
	Safety In	0.01	0.01	0.08	0.99	0.32

## Data Availability

Data available on request.

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
