# Peer review of "Psychological Reactions during and after a Lockdown: Self-Efficacy as a Protective Factor of Mental Health"

_ijerph, 2023, doi:10.3390/ijerph20176679_

Round 1

Reviewer 1 Report

Use a positive tone in the title.

Include numerical values in the abstract and revise survey dates to be consistent throughout the text.

Revise the references so that authors who appear in the text as background to the confinement should have published research after the pandemic.

Revise the introduction as it has a general aim at the beginning and two aims at the end that need to be revised. The objectives should be in the infinitive, with active verbs, transitive and subject to only one interpretation. Hypotheses are mixed with procedures and results. They should clearly express the relationship or difference in terms of expected results and relate to the research problem.

The conclusion should answer the questions of the study (perhaps not because of lack of clarity in the aims and hypotheses).

The wording is correct, although there are errors in the word COVID-19, which should always be capitalised. The instruments used should all be written in the same structure: full name (acronym) and explanation.

The design used is not explained. I believe the sampling is not random, but this is not explained. The data collection instruments are adequate, but the validity and reliability of each instrument is not described. The data collection procedure should be explained in the methods section and the steps taken in the research should be made clear. Unify the dates of the surveys and use the same dates in all sections. Explain why the first survey was completed in one week and the second in six months.

If the aims and hypotheses section and the design are clear, the data analysis and selection of results will be clearer. Include a short and clear title in the tables and the explanation can be included in the text.

The discussion will be improved by clearly stating the aims at the end of the introduction.

Include a discussion of the limitations of the study and how they may affect the conclusions. Lack of suggestions for future research.

Some conclusions appear in the discussion and the conclusion section does not clearly state the conclusions according to the objectives. If the objectives are clear, as many conclusions as objectives can be drawn.

References are up to date, although some data refer to the period of the pandemic and the references are before 2020. For example, reference 72.

Some referencing errors should be corrected: In the text, references are sometimes written in APA: (WHO, 2020); (Faul, Erdfelder, Buchner, & Lang, 2009); (Iachini et al., 2021). In the text for multiple authors, the first author et al. and the reference are used: Di Giorgio et al. (70); Schoenfeld et al. (85); Benassi et al. (71); Poole et al. (91). Check the year of some references which are different in the list and in the text: (70) 2020 or 2021; (91) 2017 or 2018. The reference in the list Schwarzer, R. & Jerusalem, M. (1995) is not spelled correctly. Remove self-citation 38. High level of plagiarism 36% and 12% of self-citation

Author Response

We thank the reviewer for her/his work and for the comments on our manuscript. We hope we have satisfactorily answered all the issues raised.

Point 1: Use a positive tone in the title.

Response 1: Ok, now the title is “Psychological reactions during and after a lockdown: self-efficacy as a protective factor of mental health”.

Point 2: Include numerical values in the abstract and revise survey dates to be consistent throughout the text.

Response 2: As suggested by the reviewer, we added numerical values in the abstract (i.e., dates, sample composition, proportion of the variance explained by the independent variable in the regression model and average values of measures taken during and after lockdown). Besides, we made the survey dates consistent throughout the text.

Point 3: Revise the references so that authors who appear in the text as background to the confinement should have published research after the pandemic.

Response 3: We have revised the references that appear in the Introduction as background to the confinement. When introducing the effects of confinement related to COVID-19 we have left the studies published after the pandemic (i.e. 2021/2023) (please see lines 40-11; also 68-74), and removed those published before 2020 (i.e. before the pandemic started) and limited to certain parts of the manuscript, and with appropriate justification, those published during 2020 (i.e. during the pandemic and lockdown, see lines 52-54 and 81-82).

As a matter of fact, we left the references published before the pandemic when the effects of confinement related to causes other than COVID-19 were illustrated (e.g., see lines 46-49) or served to justify the choice of certain predictors (see lines 100-101).

Point 4: (i) Revise the introduction as it has a general aim at the beginning and two aims at the end that need to be revised. The objectives should be in the infinitive, with active verbs, transitive and subject to only one interpretation. (ii) Hypotheses are mixed with procedures and results. They should clearly express the relationship or difference in terms of expected results and relate to the research problem.

Response 4: (i) We revised the Introduction and placed the revised aims of the study at the end of the section after the research problems (see lines: 92-97). Furthermore, we took care that the objectives had active, transitive and infinitive verbs and were subject to only one interpretation. Now it is: “Therefore, the aim of the current study was: (i) to investigate which psychological and situational factors predicted the perceived stress and anxiety among individuals subjected to the lockdown to counter the COVID-19 outbreak in Italy; (ii) to review the mental health status of individuals once the restrictive measures were over; (iii) to assess the role of self-efficacy in predicting mental health.

More importantly, immediately after the objectives, we included a hint of the procedures for achieving them (see lines: 97-106) and the hypotheses (see lines: 107-126).

(ii) As the reviewer suggested, now the hypotheses are clearly separated from procedures and results. Besides, we took care that the hypotheses clearly referred to the research problem and expressed the expected results (see lines: 107-126). Now it is: “Based on previous studies [1-10; 15-26], we put forward the following hypotheses: H1 A significant relationship between psychological and situational factors and state anxiety levels should emerge. Specifically, higher levels of state anxiety would be predicted by being a woman, young and single, having children at home and living in a city. Moreover, we expected higher levels of state anxiety to be associated with higher feeling of isolation and trait anxiety, a lower sense of security, more outside than inside the home, a reduced number of outings and a longer duration of lockdown; H2 If perceived stress is predicted by the same factors as anxiety, then we would expect the same associations as in H1. Importantly, the analysis will also allow to identify common predictors of anxiety and stress; H3 If the lockdown results in high levels of stress and anxiety, an increased sense of isolation and a reduced sense of security outside the home, then after the lock-down a decrease in state anxiety, stress and sense of isolation and an increased sense of security outside the home should be observed. However, the overall picture is likely to be worse for women than for men; H4 If self-efficacy plays a protective role to recover mental well-being after a lockdown, then a significant relationship between self-efficacy and stress, anxiety, feelings of isolation and security should be observed. Specifically, higher levels of self-efficacy should be associated with lower levels of stress, state anxiety and isolation, and higher levels of safety outside the home.

Point 5: The conclusion should answer the questions of the study (perhaps not because of lack of clarity in the aims and hypotheses).

Response 5: Ok. We have rewritten the conclusions so that they answer more directly and explicitly to the questions, objectives and hypotheses of the study (see lines: 474-506). Now it is: “The adoption of a lockdown over the centuries has proven to be an effective measure not only to stem the spread of disease and the emergence of pandemics (see the recent COVID-19), but, in general, to deal with perceived emergency situations. However, there may be negative consequences on the mental health of individuals [2-11]. In line with this, the current study revealed that:

(i)           being female, higher levels of trait anxiety and feelings of isolation were associated with higher levels of perceived stress and state anxiety during lockdown. Interestingly, the presence of children at home was associated with higher levels of anxiety but not stress, whereas young age was asso-ciated with higher levels of stress but not anxiety;

More importantly, the novelty of this study lies in having compared the picture that emerged during confinement with the end of restrictions:

(ii)         after the lockdown, levels of stress, state anxiety and feelings of isolation had decreased and safety outside the home had increased; but this was more true for men than for women;

Finally, the current study crucially showed that:

(iii)        self-efficacy may play a protective role to recover mental well-being after a lockdown. In fact, higher levels of self-efficacy were associated with lower levels of perceived stress, state anxiety and sense of isolation and with higher levels of perceived safety outside home.

In sum, the results of this study contribute to inform about the common and different predictors of anxiety and stress and which segments of the population are most affect-ed, both during and after lockdown.  This information should be used to implement the lockdown by trying to mitigate its negative consequences especially on women and young people. In this respect, the current study emphasizes the key role played by self-efficacy. Specifically, it would be important that government decision-makers adopt measures and/or communication strategies that support the individuals’ perceived self-efficacy. Therefore, future studies should investigate and test forms of communi-cation and activities to be proposed to individuals during a lockdown to increase their self-efficacy. Furthermore, further studies should investigate whether self-efficacy works as protective factor also in the more economically disadvantaged portion of the population [9] and/or have unsatisfactory housing solutions for confinement [97].”

Point 6: The wording is correct, although there are errors in the word COVID-19, which should always be capitalised. The instruments used should all be written in the same structure: full name (acronym) and explanation.

Response 6: Ok. Now the word COVID-19 is always capitalized. Moreover, the standardized instruments are written in the same structure, i.e. full name (acronym) and explanation.

Point 7: (i) The design used is not explained. I believe the sampling is not random, but this is not explained. The data collection procedure should be explained in the methods section and the steps taken in the research should be made clear. (ii) The data collection instruments are adequate, but the validity and reliability of each instrument is not described. (iii) Unify the dates of the surveys and use the same dates in all sections. Explain why the first survey was completed in one week and the second in six months.

Response 7: (i) As regards the sampling/recruiting method and the data collection procedure we now specify that “Participants in the first survey were recruited through social media announcements on the Internet, word of mouth and e-mail lists from the laboratories of the universities involved. Informed consent was obtained from all participants. The link to participate in the online survey was directly available in the posts and emails sent out. At the end of the survey, participants were asked for permission to be contacted again for possible follow-up of the study.” (lines: 141-146). Besides, in the procedure section we now specify: “In both the first and second survey, once participants clicked on the link received by e-mail or found on social media on the Internet, they visualized the informed consent page and the instructions. …..” (see page: 205-207). Finally, as regards the “steps taken in the research” we now specify more clearly in the 'participants' section when recruitment took place, the specific dates of the first and second phases, and what participants did once they received the survey link;

(ii) The validity and reliability of the STAI, PSS and GSES scale is now indicated (please see lines: 172-174, 183-185, 190-192);

(iii) the dates of the surveys have been unified and the same dates have been reported in all sections. Importantly, now we explain the temporal difference between the first and second survey. Specifically, it is stated that the duration of data collection during the first survey was constrained by reaching a sufficient number of participants for an adequate power of statistical testing. Therefore, data collection ended as soon as the minimum number of participants indicated by GPower had been exceeded. We now state: “Data collection was carried out during the first lockdown imposed in Italy to counter the spread of COVID-19. It began on April 23rd 2020 and ended on May 2nd 2020, when the minimum number of participants indicated by G*Power had been exceeded.(see lines: 133-135). As regards the second survey we state (see lines: 458-462): “In contrast to the high level of cooperation in the first survey, many respondents were no longer available in the second survey and others, having returned to their pre-pandemic lives, had less time to complete the questionnaires. This resulted in high drop-out rates and longer data collection times in the second survey than in the first.

Point 8: If the aims and hypotheses section and the design are clear, the data analysis and selection of results will be clearer. Include a short and clear title in the tables and the explanation can be included in the text.

Response 8: A data analysis section has been added (see paragraph 2.4. in the Method section). In addition, we have reorganized the results section so that the results are directly linked to the relevant hypotheses. Finally, a short, and hopefully, clear title has been included in each table and the explanation of the table is in the text.

Point 9: The discussion will be improved by clearly stating the aims at the end of the introduction.

Include a discussion of the limitations of the study and how they may affect the conclusions. Lack of suggestions for future research.

Response 9: As suggested by the reviewer, the aim of the study presented at the end of the introduction is now repeated at the beginning of the discussions. Furthermore, we have reorganized the discussions so that each sub-section refers to both specific objectives and results sections, which in turn respond to the specific hypotheses. Finally, we included a paragraph on the limitations of the study in the discussions (see lines: 458-473) and added suggestions for future research in the conclusions  (see lines: 501-507).

Point 10: Some conclusions appear in the discussion and the conclusion section does not clearly state the conclusions according to the objectives. If the objectives are clear, as many conclusions as objectives can be drawn.

Response 10: The reviewer is right. Accordingly, we have removed some conclusions from the discussion and now the conclusion section is more clearly related to the objectives.

Point 11:  References are up to date, although some data refer to the period of the pandemic and the references are before 2020. For example, reference 72.

Response 11: Please see Response 3. In short, some past studies (i.e. pre-COVID-19) have been maintained because they serve to interpret/discuss the data of the current study in the light of evidence from other studies that have dealt with mental health and emergency situations. This is precisely the case with the reference mentioned by the reviewer:           “Sprang, G. & Silman, M. Posttraumatic stress disorder in parents and youth after health-related disasters. Disaster Med Public 2013, 7(1), 105-110.” 

Point 12: Some referencing errors should be corrected: In the text, references are sometimes written in APA: (WHO, 2020); (Faul, Erdfelder, Buchner, & Lang, 2009); (Iachini et al., 2021). In the text for multiple authors, the first author et al. and the reference are used: Di Giorgio et al. (70); Schoenfeld et al. (85); Benassi et al. (71); Poole et al. (91). Check the year of some references which are different in the list and in the text: (70) 2020 or 2021; (91) 2017 or 2018. The reference in the list Schwarzer, R. & Jerusalem, M. (1995) is not spelled correctly. Remove self-citation 38. High level of plagiarism 36% and 12% of self-citation

Response 12: As suggested by the reviewer referencing errors have been corrected. However, we maintained the reference 38 (now 41, i.e. Iachini et al., 2021) because it is used to justify the effect size to calculate the sample size for the current study and for other methodological details.

The level of plagiarism detected by the software made available by the University of Campania (https://www.ithenticate.com/) on the revised version is around 3% and only reflects phrases or formulas commonly used in a research paper and not intellectual property of any kind (e.g., “the impact of lockdown on mental health”, “….was associated with higher level of anxiety….”, “levels of self-efficacy…..”, “previous studies that reported higher scores in stress, anxiety, and depressive symptoms in younger people compared to older ages”….). Where small phrases from other articles have been used (a couple of cases in total), we have used quotation marks and mentioned author and page number (the latter where available). For clarity, by including the methodological part, with the standard description of materials and analysis models, the level of plagiarism rises to 7% (but we would like to stress again that these are small standard phrases or definitions found in many other research works).

Finally, out of 97 references only 2 came from our COVID research group (by the way, they were necessary to justify some methodological choices and propose future studies). Specifically we refer to:

  1. Iachini, T.; Frassinetti, F.; Ruotolo, F.; Sbordone, F. L.; Ferrara, A.; Arioli, M.; et al. Social distance during the COVID-19 pandemic reflects perceived rather than actual risk. Int J Environ Res Public Health 2021, 18(11), 5504.
  2. Fornara, F., Mosca, O., Bosco, A., Caffò, A.O., Lopez, A., Iachini, T., Ruggiero, G., Ruotolo, F., Sbordone, F.L., Ferrara, A., Cattaneo, Z. Space at home and psychological distress during the Covid-19 lockdown in Italy. J Environ Psychol, 2022, 79, 101747.

Reviewer 2 Report

Thank you for submitting your manuscript for review. Overall I feel the rationale is clear, and your paper is well written. I do wonder whether parts of the discussion might help strengthen and contextualise the introduction and rationale, in particular the parts about humans being deeply social.

It is a shame that such a small subset of the original sample participated in the follow up study, however, although the drop out rate is high, this still seems generally like a decent sample size. The initial sample is great – there is a really good balance of males and females, and a good age range of participants, reflecting the cooperative nature of data collection.

Generally I feel the results are reported clearly, though in places I feel this could be clarified (e.g. in 3.1.1 I think it would be clearer to say something like “higher feelings of isolation, higher trait anxiety, and being female all predicted higher state anxiety whereas feeling safer outside of the home predicted lower state anxiety)”. It may also be more helpful to exclude the p values from this table, given they are all so low, and just use * to indicate p<.001. This is also suggested for section 3.1.2.

I think it would be helpful to include the interaction plots for the significant interaction effects to help illustrate the findings. I would suggest including a table of descriptive statistics for section 3.2. because it can be difficult to follow all of the subsections in a meaningful way, and there are some difficult statements such as “Moreover, stress levels reported during (M= 18.84; SE= .63) tended to be higher than those reported after confinement (M= 17.63; SE= .57) (F(1, 172)= 3.73, p= .055, η2p = .02)” even though this isn’t significant. There is an earlier statement of non-significance in section 3.2.1 (p=.053) and I think it would be on par to refer to this finding as non-significant too. Nevertheless, the findings are interesting and close to the threshold therefore referring to them in a table I think would be more transparent.

Fine - some minor typographical errors. 

Author Response

Response to Reviewer 2 Comments

We thank the reviewer for agreeing to review our work and for the helpful advice and issues raised. We hope we have provided clear and satisfactory answers.

Point 1: Thank you for submitting your manuscript for review. Overall I feel the rationale is clear, and your paper is well written. I do wonder whether parts of the discussion might help strengthen and contextualise the introduction and rationale, in particular the parts about humans being deeply social.

Response 1: As suggested, we have moved the parts about “humans being deeply social” to the beginning of the introduction (see lines: 55-62). We believe this helps to strengthen the idea of the study.

Point 2: It is a shame that such a small subset of the original sample participated in the follow up study, however, although the drop out rate is high, this still seems generally like a decent sample size. The initial sample is great – there is a really good balance of males and females, and a good age range of participants, reflecting the cooperative nature of data collection.

Response 2: We thank the reviewer for this comment. Importantly, we used part of her/his commentary in the “limitations” section of the study, especially about the high level of cooperation that led to high levels of participation in the first survey. See page 1 lines 459-462: “In contrast to the high level of cooperation in the first survey, many respondents were no longer available in the second survey and others, having returned to their pre-pandemic lives, had less time to complete the questionnaires. This resulted in high drop-out rates and longer data collection times in the second survey than in the first.”

Point 3: Generally I feel the results are reported clearly, though in places I feel this could be clarified (e.g. in 3.1.1 I think it would be clearer to say something like “higher feelings of isolation, higher trait anxiety, and being female all predicted higher state anxiety whereas feeling safer outside of the home predicted lower state anxiety)”. It may also be more helpful to exclude the p values from this table, given they are all so low, and just use * to indicate p<.001. This is also suggested for section 3.1.2.

Response 3: We agree with the reviewer and consequently rewrote the results part (for both anxiety and stress) following her/his advice. We did the same for the p-values in the tables.

Point 4: I think it would be helpful to include the interaction plots for the significant interaction effects to help illustrate the findings. I would suggest including a table of descriptive statistics for section 3.2. because it can be difficult to follow all of the subsections in a meaningful way, and there are some difficult statements such as “Moreover, stress levels reported during (M= 18.84; SE= .63) tended to be higher than those reported after confinement (M= 17.63; SE= .57) (F(1, 172)= 3.73, p= .055, η2p = .02)” even though this isn’t significant. There is an earlier statement of non-significance in section 3.2.1 (p=.053) and I think it would be on par to refer to this finding as non-significant too. Nevertheless, the findings are interesting and close to the threshold therefore referring to them in a table I think would be more transparent.

Response 4: As suggested by the reviewer plots have been added for the significant interactions. In addition, a table with descriptive statistics has been added (see Table 3). Finally, as reviewer suggested we removed the effects approaching statistical significance from the results and included them in the table. Finally, as suggested by the reviewer, we removed the effect tending towards significance from the results (i.e. that of the interaction between gender and type of anxiety). Instead, we left the Before-After effect related to stress, making it clear both in the results (lines: 284-285) and in the discussion (lines: 434-435) that this is an effect that tends towards statistical significance. 
